# Global landscape analysis of no-fault compensation programmes for vaccine injuries: A review and survey of implementing countries

Randy G. Mungwira[ID][1¤]*, Christine Guillard[2], Adiela Saldaña[3], Nobuhiko Okabe[4], Helen Petousis-Harris[5], Edinam Agbenu[6], Lance Rodewald[7], Patrick L. F. Zuber[2]

1 Department of Molecular Medicine and Development, University of Siena, Siena, Italy, 2 Access to Medicines and Health Products Division, World Health Organization, Geneva, Switzerland, 3 Instituto de Salud Pública de Chile, Santiago, Chile, 4 Kawasaki City Institute for Public Health, Kawasaki-City, Japan, 5 University of Auckland, Auckland, New Zealand, 6 World Health Organization, Ouagadougou, Burkina Faso, 7 Chinese Center for Disease Control and Prevention, Beijing, China

¤ Current address: World Health Organization Malawi Country Office, Lilongwe, Malawi
* rgmungwira@gmail.com

**Data Availability Statement:** All relevant data are within the manuscript and its Supporting Information files.

## Abstract

To update the landscape analysis of vaccine injuries no-fault compensation programmes, we conducted a scoping review and a survey of World Health Organization Member States. We describe the characteristics of existing no-fault compensation systems during 2018 based on six common programme elements. No-fault compensation systems for vaccine injuries have been developed in a few high-income countries for more than 50 years. Twenty-five jurisdictions were identified with no-fault compensation programmes, of which two were recently implemented in a low- and a lower-middle-income country. The no-fault compensation programmes in most jurisdictions are implemented at the central or federal government level and are government funded. Eligibility criteria for vaccine injury compensation vary considerably across the evaluated programmes. Notably, most programmes cover injuries arising from vaccines that are registered in the country and are recommended by authorities for routine use in children, pregnant women, adults (e.g. influenza vaccines) and for special indications. A claim process is initiated once the injured party or their legal representative files for compensation with a special administrative body in most programmes. All no-fault compensation programmes reviewed require standard of proof showing a causal association between vaccination and injury. Once a final decision has been reached, claimants are compensated with either: lump-sums; amounts calculated based on medical care costs and expenses, loss of earnings or earning capacity; or monetary compensation calculated based on pain and suffering, emotional distress, permanent impairment or loss of function; or combination of those. In most jurisdictions, vaccine injury claimants have the right to seek damages either through civil litigation or from a compensation scheme but not both simultaneously. Data from this report provide an empirical basis on which global guidance for implementing such schemes could be developed.

**Funding:** The author(s) received no specific funding for this work.

**Competing interests:** The authors have declared that no competing interests exist.

## Introduction

No-fault vaccine injury compensation programmes are established to compensate individuals who experience a rare vaccine-related injury due to the inherent risk of vaccination (e.g. intussusception in an infant following vaccination with a well manufactured and administered rotavirus vaccine, or a life-threatening anaphylactic reaction following any vaccine) [1–3]. These programmes do not require the injured party or their legal representative to prove negligence or fault by the vaccine provider, health care system or the manufacturer prior to compensation. They serve to waive the need for accessing compensation through litigation processes, which are often viewed as an adversarial approach requiring establishment of fault by at least one party prior to compensation [2]. The term ¨no-fault¨ implies a measure put in place by public health authorities, private insurance companies, manufacturers and other stakeholders to compensate individuals inadvertently harmed by vaccines [4]. In 1961, Germany was the first country to implement a no-fault compensation programme that covered vaccine injuries [2]. This stemmed from the 1953 supreme court ruling to compensate people injured with compulsory smallpox vaccination [2]. The drive to implement no-fault compensation programmes in most jurisdictions increased with reports of adverse events following immunisation with diphtheria-tetanus-whole cell pertussis in the 1970s [2].

However, with continued improvements in reporting and investigation of vaccine safety events, including in low- and middle-income settings, WHO Member States are identifying and documenting events that have scientific evidence of causal association to vaccination [5]. This is accompanied by increasing interest for national no-fault compensation policies related to vaccine injuries [6–9]. As of 2010, compensation schemes for vaccine-related injuries had been identified and characterized in nineteen out of WHO's 194 Member States [2]. At the time, these programmes were exclusively implemented in high-income countries. Previous reviews have described the characteristics of existing programmes based on the six common elements identified by Evans in 1999 including administration and funding, eligibility, process and decision making, a standard of proof, elements of compensation, and litigation rights [1, 2]. We conducted a global survey of the status of vaccine injury no-fault compensation programmes (complemented by triangulation of information from multiples sources) with the aim to update the inventory of such programmes and evaluate and update their characteristics to forecast the next segment of adopters and guide policy formulation.

## Materials and methods

Initially, a landscape analysis and scoping review of published and unpublished literature were conducted to update the inventory of countries that have implemented vaccine injury no-fault compensation programmes (scoping review protocol not registered). Published data was supplemented with official documents accessed from government websites (where available). Structured literature search was done using PubMed, Excerpta Medica dataBASE (EMBASE), Cumulative index to Nursing and Allied Health Literature (CINAHL) and Global Online Access to Legal Information (GOALI) using the following predefined keywords: vaccine injury AND compensation programs; AEFI AND compensation; vaccine AND injury AND no-fault compensation; vaccine damage payment; and vaccine liability claims (S1 File). Using a lower cutoff period of 31 Dec 2009 to supplement on previous reviews, 41 articles published in English with relevant information were reviewed (Fig 1). This descriptive analysis of the characteristics of WHO member states with no-fault compensation programmes implemented is published elsewhere [10].

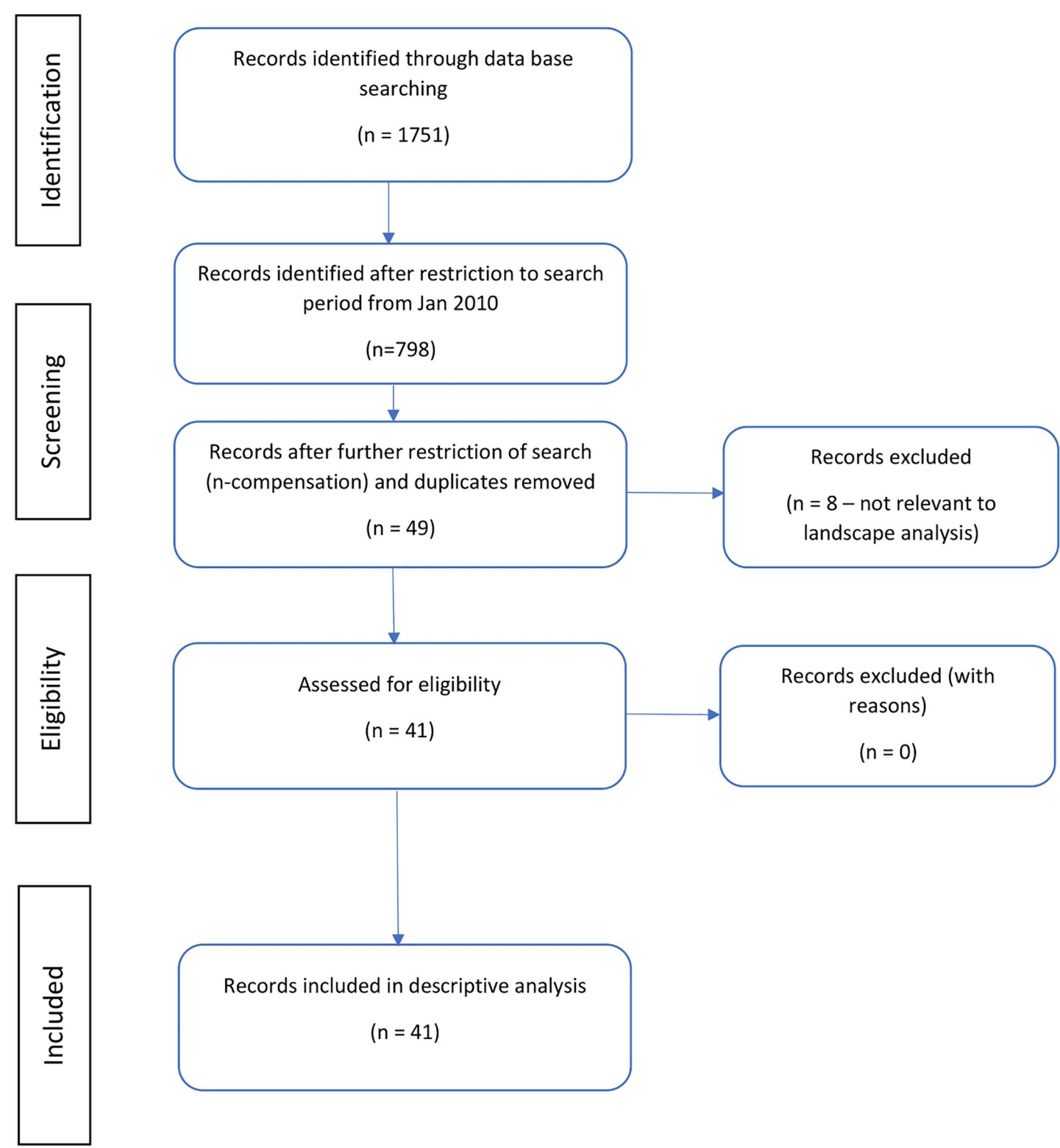

**Fig 1. PRISMA flow diagram indicating structured literature search to address a descriptive analysis of current policies and practices of no-fault compensation programmes for vaccine injuries.**

In addition, all WHO Member States were approached, and screening was done using several methods to identify those with a no-fault compensation programme for vaccine injuries. We approached several professional networks including immunization programme focal points in WHO Regional or Country Offices and local Ministry of Health in Member States. Screening for programmes was also conducted amongst current and past members of the Global Advisory Committee on Vaccine Safety (GACVS) [11], and conference attendants (Global Immunization Meeting [12], Vaccine Safety Net meeting and International Conference of Drug Regulatory Authorities [13]). During the same period, the WHO Immunization, Vaccines and Biologicals Department repository and a global survey of national immunization technical advisory groups collected information on the presence of systematic compensation programmes for vaccine injuries. This data was used to triangulate the presence or absence of programmes for compensating vaccine injuries. For each country with a no-fault compensation programme for vaccine injuries identified through our approach, an expert with in-depth knowledge of the no-fault scheme was identified through colleagues in WHO country offices or National Immunization Programme Focal Points within the Member States, and invited to complete a structured online survey (S4 File). The questionnaire was created using a data collection tool which is based on LimeSurvey (Version 2.06+ Build 151215) to collect data on the structure, perceived benefits and operational challenges of existing programmes. The survey was designed by the Global Vaccine Safety team at the WHO Headquarters in Geneva, Switzerland. It was piloted and further refined before being administered. An independent scientific committee consisting of selected members of the GACVS, and additional immunization experts validated the survey and oversaw the conduct of the study to ensure scientific rigor. An email was sent out to participants (S2 File) inviting them to complete the online survey and responses were received from 03 July to 31 September 2018. To ensure data accuracy, survey respondents were encouraged to submit supporting documents. The survey tool was also made available in French, an official WHO language (S5 File).

A scientific review of the protocol was conducted in collaboration with the academic and scientific committee from the University of Siena, Master of Vaccinology and Pharmaceutical Clinical Development programme. The study was granted exemption from full ethical review by the WHO Ethics Committee since the study involved human subjects participating in their professional capacity (as staff or affiliates of WHO regional or country offices or Ministry of Health) and sharing information available in the public domain. Informed consent was sought from all participants prior to collecting any study-related data.

## Results

All 194 WHO member states were screened for the presence of no-fault compensation programs for vaccine injuries. We received feedback from 151 countries who responded to an initial screening step to determine the presence of a no-fault compensation programme. From these responses, we identified 25 member-states implementing no-fault compensation programmes (Fig 2) that met the predefined definition [10]. Through the survey and other data sources (i.e. government documents where available) we evaluated 23 existing programmes (including two from Japan) based on the six common elements reported in previous reviews [1, 2]. Regional distribution and characteristics of implementing countries are described separately [10].

The number of countries implementing no-fault compensation programmes for vaccine injuries has increased steadily from 19 in 2010 to 25 in 2018. As compared to previous decades there is, however, no acceleration in the number of countries. In recent years and for the first

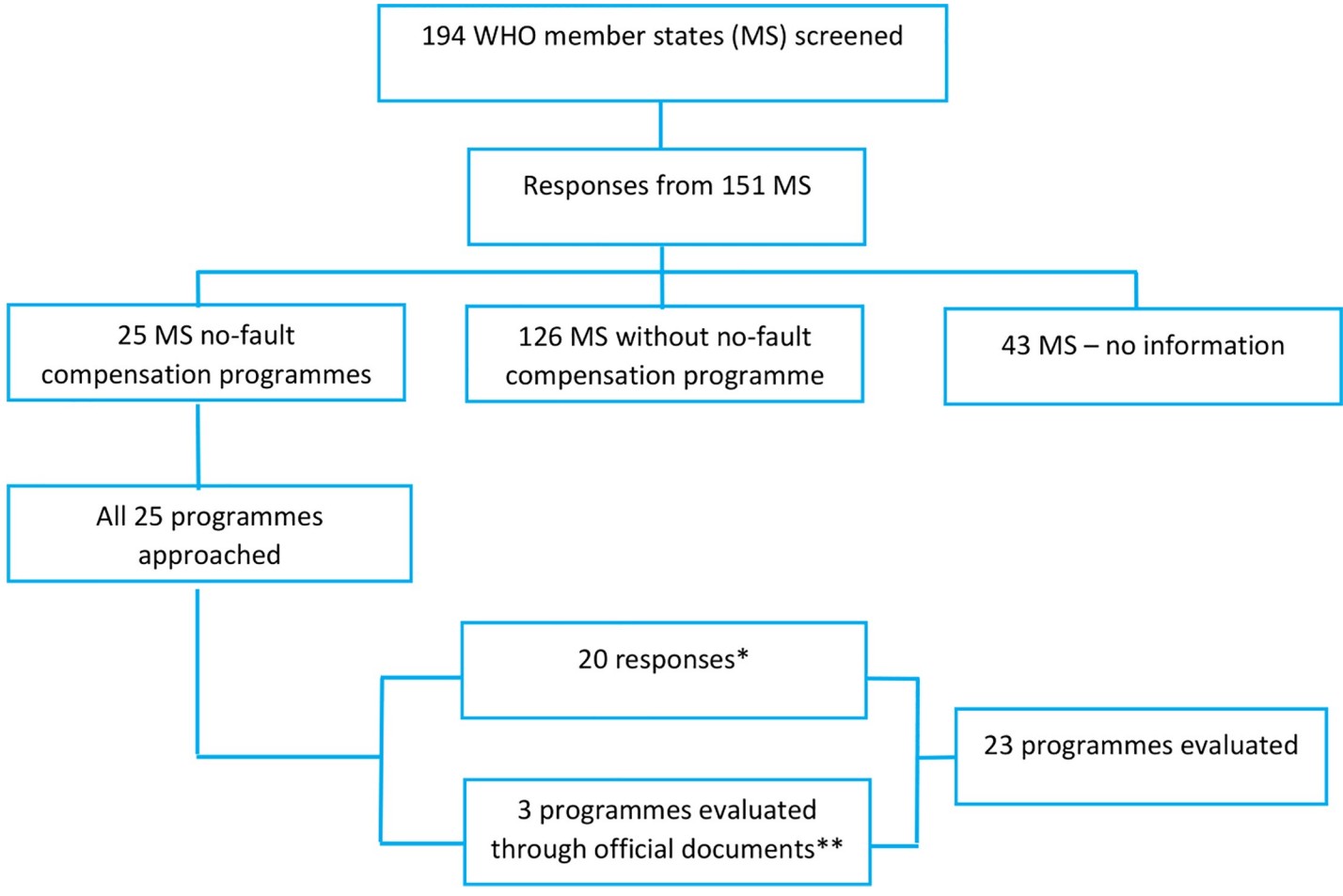

**Fig 2. Member States screened for existence of vaccine injuries no-fault compensation programmes and number of programmes evaluated.** * 19 countries responded to survey; Japan provided information for two programmes ** Latvia, Nepal and Viet Nam.

time, a low and a lower-middle-income country, Nepal and Viet Nam respectively, have instituted such programmes [14, 15]. Table 1.

## Administration and funding

**Administration.** Fifteen (65%) of the no-fault compensation programmes for vaccine injuries are administered at the central government level. Germany, Italy, Republic of China

**Table 1. No-fault compensation programme for vaccine injuries distributed by countries and continents.**

| Continent | Number of countries | Countries |
|---|---|---|
| Africa | 0 | None |
| America | 2 | United States, Canada |
| Asia | 6 | China, Japan, South Korea, Viet Nam, Nepal, Thailand |
| Europe | 16 | Austria, Denmark, Finland, France, Germany, Hungary, Iceland, Italy, Luxembourg, Norway, Russia, Latvia, Slovenia, Sweden, Switzerland and United Kingdom |
| Oceania | 1 | New Zealand |

and the Province of Quebec in Canada are the only jurisdictions implementing the compensation programme at the province level (17%). Finland and Sweden are the only countries where programmes are administered by the insurance sector [16].

Since its establishment in 1970, the programme in Switzerland was administered at the cantonal level (each of 26 states that compose the confederation). In 2016, the Swiss compensation policy was amended, and the administration of the programme is done by the central government. In Italy, the programme was decentralized in 2001 to be administered at province level in regions with the ordinary statute but remained run by the central government in regions with special statute. In 2014, the programme in the People's Republic of China was amended requiring all 31 provinces to implement compensation mechanisms for vaccine injuries [4]. Administration of the Chinese programme involves all levels of government: filing of claims and causality assessment of events is done at district or county level; operational procedures for compensation are set at province level and general vaccine injury compensation policies including definitions of what constitutes a vaccine injury are determined at the central government level. The programme in Japan is also implemented at all levels of government.

**Funding.**   Fifteen (65%) of the programmes are government funded including those being implemented in low- (Nepal) [14] and lower-middle-income settings (Viet Nam) [15]. The programmes in Finland and Sweden are funded by the insurance sector financed by contributions from pharmaceutical companies marketing their products in these jurisdictions. Although administered at the government level, the programme in Norway is also funded by a special insurance organization, the Drug Liability Association. In Latvia, the Treatment Risk Fund is funded through contributions from medical institutions [17], hence it also acts as professional indemnity insurance. In China, Japan and the Republic of Korea, there are two different programmes covering injuries arising from vaccines listed in the national immunization programme (NIP) and non-NIP vaccines [4, 18, 19]. These programmes are funded differently, with government funding NIP vaccines, and pharmaceutical companies or market authorization holders funding non-NIP vaccine injuries. The USA programme is funded by a flat-rate tax of 0.75 USD on each disease prevented in each vaccine dose (e.g., 2.25 USD for measles mumps rubella vaccines, and 0.75 USD for *Haemophilus influenza* type B vaccine) [2, 20]. New Zealand has an Accident Compensation Corporation (ACC) which compensates for vaccine injuries under a general compensation for accidents and treatment injuries. The ACC is funded from the contribution of general taxation, and levies collected from employee earnings, businesses, vehicles licensing and fuel [21].

## Eligibility

**Vaccines.**   Thirteen programmes (57%) compensate for injuries arising from registered and recommended vaccines for children, pregnant women or adults (e.g. influenza vaccines) and for special indication (e.g. travel or occupation) within the jurisdiction. Five (22%) of the programmes cover injuries arising from mandatory or vaccines pro-actively recommend by law only including in France, Hungary, Italy, Slovenia and Japan (for injuries arising from NIP listed vaccines category A which are administered to achieve basic herd immunity). The programmes in the United Kingdom and Province of Quebec in Canada [22] cover for injuries arising from vaccines against specific diseases of infections as listed in their legislation.

**Timelines of injury and vaccination.**   Timelines vary considerably from programme to programme. In the United Kingdom, claims can only be filed when the child is two years old. For adults, whichever is the latest of the following dates: either on or before their 21st birthday (or if they have died, the date they would have reached 21 years old), or within 6 years of vaccination. In the USA, the Province of Quebec, Denmark, Italy and Norway, the programmes

compensate for injuries that occur within three years of vaccination or initial appearance of symptoms of the vaccine injuries. In case of death, the USA programme compensates if it occurs within two years of vaccination and not more than four years from the initial date of symptoms of the vaccine injury that led to death. In Denmark and Norway, the maximum interval between the occurrence of vaccine injury and filing a claim is 10 and 20 years respectively. In Switzerland, claims can be submitted up to when one is 21 years old for childhood vaccines or within five years of vaccination. Similarly, the programmes in Japan (for non-NIP vaccines) and the Republic of Korea have a five years window for filing claims. Finland and France have a 10 years window for filing a claim, in China, this varies by province. The programmes in Austria, German, Hungary, Japan (NIP vaccines), Luxembourg, Slovenia, Sweden, and New Zealand do not have specified timelines between the occurrence of a vaccine injury and filing a claim.

**Injured party.**   Fifteen programmes (65%) compensate all individuals who experience an eligible injury arising from a vaccine administered within their jurisdiction. In Denmark, Slovenia and China, only citizens are eligible for compensation. Whilst in the Province of Quebec in Canada, Germany, Italy and Japan (programme for NIP listed vaccines), only province residents who experience a vaccine injury are eligible for compensation.

**Types of injuries covered.**   All countries implementing no-fault compensation programmes have a threshold of eligibility for vaccine injury and these include: injuries resulting in financial loss or permanent or significant injury (i.e. medical disability), serious health damage or death, severe injuries exceeding normal post-vaccination reactions, severe disability secondary to vaccination against a specified disease in the legislation, serious adverse events following immunization (AEFI) or disability as per predefined criteria. In the Republic of Korea, compensation may be considered for any vaccine injury whose treatment cost beyond USD 260 (300,000 Korean Won). Although the schemes studied are primarily designed to compensate for inherent risks of vaccination ("no-fault") Injuries arising from negligence (i.e. vaccine quality defects or immunization errors) are also covered under the schemes of twelve of the 23 programmes (52%) studied. In the remaining jurisdictions, injuries arising from negligence are handled separately either under a medical malpractice indemnity cover or through civil litigation.

## Process and decision making

**Process.**   In all the compensation programmes, the process is initiated by the injured party or their legal representative filing a claim with a special administrative unit handling vaccine injury compensation. In New Zealand, this process is initiated by the healthcare worker reviewing the injury, who then notifies the ACC. Eighteen of the programmes (78%) are purely administrative in nature with a unit consisting of health officials or an insurance organization that processes claims operating under a pre-set legislation. Five of the programmes (22%) in Austria, Finland, Hungary, and the USA have an approach that either combines both administrative and civil litigation processes or are considered a judicial review in Denmark. The national vaccine injury compensation programme in the USA involves a special court that deliberates on claims and makes the final decision on compensation for injuries pre-listed, or upon examination of an expert witness for non-listed injuries, an approach like civil litigation.

**Decision making.**   In the purely administrative programmes, a group of medical experts reviews individual cases of vaccine injuries filed for compensation and make the decision based on available evidence. Once a decision to compensate or refuse compensation is made, the recommendations of the expert group are forwarded to the programme for action. In jurisdictions with both administrative and legal approaches, the final decision on compensation is

made by legal experts (e.g. Austria, Denmark, Hungary, USA). In Finland and Sweden, compensation decisions are based on civil (tort) liability laws. Decisions making process varies amongst programmes ranging between 10 days to five years depending on the nature and complexity of the claim.

## Standard of proof

All programmes reviewed require standard of proof showing a causal link between vaccination and injury. As described by Looker et al, most compensation programmes adopt the "balance of probabilities" approach which assumes that it is "more likely than not" that the vaccine caused the injury considering its nature, the consistency of time interval from vaccination, the existing medical evidence establishing an association between the injury and the vaccine including other supporting information available [2, 23]. In sixteen of the programmes (69%), the standard of proof is based on a causal association to vaccination based on standard causality assessment. In the rest of the programmes, the standard of proof is as determined by a selected group of experts. The USA compensates injuries that are listed on the vaccine injury table occurring within pre-defined timelines [24]. Claimants with injuries not listed on the vaccine injury table are required to prove that vaccine caused the injury by presenting necessary medical records or opinions which may include testimonies from expert witnesses. In China, the standard of proof is based on epidemiological causation and the regulation excludes from compensation injuries that are deemed coincidental to vaccination [4]. In Switzerland, the causality assessment of vaccine injuries is subjected to methodological approval by a group of experts. This group of experts is equivalent to a national immunization technical advisory group (NITAG), a group of experts that provides scientific recommendations for evidence-based immunization policy and programme decisions [22, 25]. In the Canadian Province of Quebec, the standard of proof is based on the existence or lack thereof of a probable causal link between the injury and the vaccine as determined by three independent medical experts appointed by the province, injured party and a third nominated by the initial two medical experts.

## Elements of compensation

In all programmes, once a final decision has been reached, claimants are compensated with either (or a combination of): a lump-sum of money; monetary compensation calculated based on medical care costs and expenses, loss of earnings or earning capacity; or monetary compensation calculated based on non-monetary criteria e.g. pain and suffering, emotional distress, permanent impairment or loss of function. Other benefits include disability pension, survival pension, or death benefits. In the province of Quebec, the amount of compensation is determined based on rules and regulations as prescribed in the Automobile Insurance Act and is identical to the compensation offered to victims of automobile accidents. In Viet Nam, compensation for disability arising from a vaccine injury is equivalent to 30 months base salary or calculated based on lost or reduced income with a standardized formula [15]. The programme in Switzerland offers compensation equivalent up to USD 70,000 aimed at covering costs related to vaccine injuries that are not covered by other third-party benefits [25].

In twelve of the programmes (52%), the amount of compensation is calculated on a case by case basis and the final amount paid out depends on the extent of the injury. In ten of the programmes (44%), the compensation amount is standardized. Compensation amounts also vary across existing compensation programmes, and across provinces in countries implementing decentralized compensation programs e.g. China [5].

### Litigation rights

In fifteen (65%) jurisdictions, claimants are required to file a vaccine injury claim with the compensation programme, but still maintain the right to pursue civil litigation against the vaccine manufacturer or health care professionals if they can prove there was a fault (i.e. vaccine quality defect). In Canadian Province of Quebec, Denmark, Hungary, New Zealand, Slovenia and Sweden, vaccine injury claims can only be filed with the compensation programme (26%).

In the USA, claimants forego their right to file for a civil claim once they have accepted compensation from the national Vaccine Injury Compensation Programme (VICP) [2]. The characteristics of the existing programmes are summarised in Table 2 below.

### Benefits of vaccine injury compensation programmes

The benefits most referred to in existing no-fault compensation programmes were: fair compensation for individuals inadvertently injured by a vaccine meant for public good and increasing confidence in public vaccination programmes. Most respondents did not consider sustenance of vaccine supply, protecting manufacturers from liability and stabilization of vaccine prices as benefits of their programmes (Fig 3). Compensation programmes were seen by respondents to enhance the legal basis of mandatory vaccination systems (in member states where such laws existed) and a sign of the government's commitment towards immunization programmes.

### Challenges of vaccine injury compensation programmes

The most notable operational challenge of the existing programmes noted by respondents was lack of public awareness of programme existence, strict requirements for standard of proof that vaccine caused injury, and long timelines for filing claims and receiving compensation (Fig 4). Despite the lack of awareness ranking as a high challenge for most programmes, few participants indicated programme accessibility as an operational challenge. Most participants did not consider their programmes to be overwhelmed by the number of claims filed. One jurisdiction cited challenges with having a non-standardized calculation of compensation amount and inadequate programme funding.

## Discussion

As countries expand vaccine use and strengthen their safety surveillance and investigative capacity, occasional severe vaccine reactions are identified [5, 26]. Subsequently, the question of fair and equitable compensation of identified vaccine injuries is more frequently raised. Previous reviews have shown that compensation programmes were perceived as interventions to offer equitable access to benefit for the injured party and lessen the financial burden for vaccine manufacturers [1, 2]. For the injured party, usually, those with adequate resources would afford to access litigation procedures creating inequity to accessing compensation [27, 28]. For vaccine manufacturers, an increase in litigation cases and substantial amounts paid out in compensation led to most players exiting the market and a subsequent significant decline in vaccine supply [29].

Proponents of this administrative approach argue it is less adversarial, more economical, and reduces the need to allot blame whilst maximizing opportunity for those with genuine vaccine injuries to access fair compensation [2, 30]. Therefore, according to such proponents, since vaccinations are usually recommended and sometimes required and enforced by global and local authorities to control infectious diseases [31], no-fault compensation programmes

**Table 2. Characteristics of existing no-fault compensation programmes for vaccine injuries.**

| VICP element | Programme attribute | Number of countries (N = 23 programmes*) |
|---|---|---|
| Admin | Central Government only | 15 (65%) |
| | Provincial Government | 3 (13%) |
| | Insurance sector | 2 (9%) |
| | Combination of the above | 3 (13%) |
| Funding source | Government only | 15 (65%) |
| | Other sources** | 8 (35%) |
| Eligibility: vaccines | Registered/recommended vaccines | 13 (57%) |
| | Mandatory vaccines | 5 (22%) |
| | Based on diseases listed in legislation | 2 (9%) |
| | Non-NIP vaccines*** | 1 (4%) |
| | No information | 2 (9%) |
| Eligibility: injured party | All injured by a vaccine administered within jurisdiction | 15 (65%) |
| | Country citizens only | 3 (13%) |
| | Province residents only | 4 (17%) |
| | No information | 1 (4%) |
| Process and decision making | Purely administrative process | 18 (78%) |
| | Combination of administrative and civil litigation processes | 5 (22%) |
| Standard of proof | Causal association to vaccination | 16 (69%) |
| | As determined by a group of experts | 5 (22%) |
| | No information | 2 (9%) |
| Compensation | Standardized compensation | 10 (44%) |
| | Case by case basis | 12 (52%) |
| | No information | 1 (4%) |
| litigation rights | Vaccine injury compensation scheme alone | 6 (26%) |
| | Both vaccine compensation schemes and tort law or civil claims are allowed◇ | 15 (65%) |
| | No information | 2 (9%) |

*22 jurisdictions evaluated with 2 programmes from Japan resulting in 23 programmes evaluated.

** Other sources include: Pharmaceutical company contribution i.e. the USA, China for non-NIP vaccine injuries, Japan for non-NIP injuries; Insurance: Finland, Norway, and Sweden have special insurance funds where all pharmaceutical companies in their jurisdiction contribute towards. France complements Gov. funding with national health insurance, Latvia has treatment risk fund.

***China, Republic of Korea, Japan—separate system for non-NIP vaccines (detailed information available only for Japan).

◇ Limited in some jurisdiction i.e. USA

are warranted and should be considered a social responsibility of each government, and global or national health authorities towards those injured by vaccines.

This evaluation has identified no-fault compensation programmes being implemented in Nepal and Viet Nam, a low and lower-middle-income country respectively. The general principles guiding the implementation of no-fault compensation programmes in these settings remains the same as identified in programmes implemented in high-income countries [1, 2]. The implementation of these programmes based on six common elements (administration and funding, eligibility, process and decision making, standard of proof, elements of compensation, and litigation rights) indicates the feasibility of developing policy guide for countries to adopt. However, the diversity of actual programme implementation across Member States supports the need for developing compensation policies adjusted to local requirements, economic capacity and legal structures. A potential limitation of our study is that, as the focus was

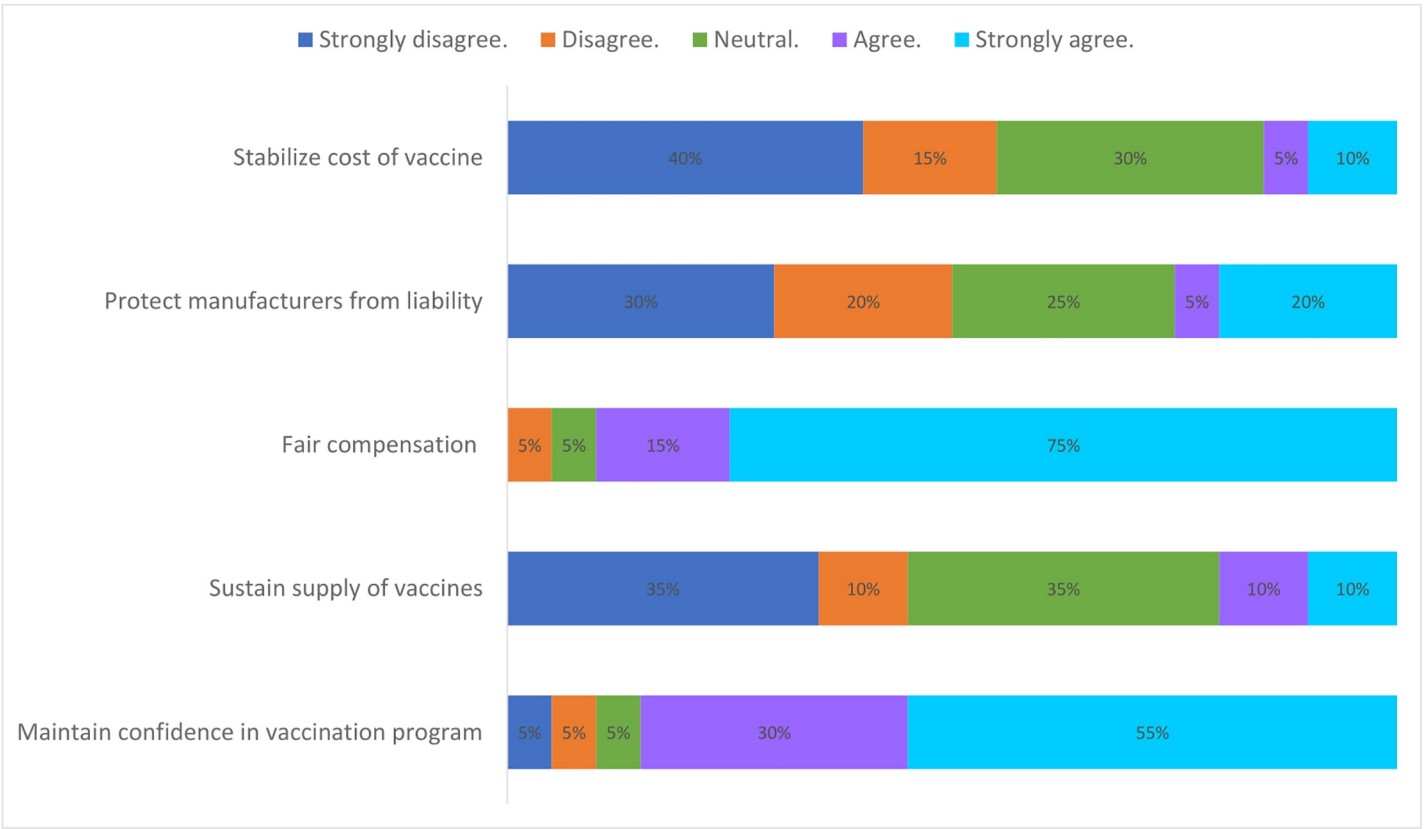

**Fig 3. Perceived benefits of no-fault compensation programmes for vaccine injuries.**

on programmes dedicated to vaccine injuries, other broader compensation mechanisms that could include vaccines–such as that from New Zealand–could have been missed.

No-fault compensation programmes are considered by some to increase adequacy and fairness of compensation as they provide clear legal guidance on how to access compensation for vaccine injuries [28]. However, implementation of such a compensation system should be considered simultaneously with the implementation of a well-established, comprehensive national social welfare system [4]. This has been thought to increase the efficiency of compensation programmes.

Advocacy for the implementation of no-fault compensation programmes should be approached cautiously to avoid distorting the public perception of vaccine safety and undermining confidence in immunization programmes. Sufficient country capacity for adverse event investigation and causality assessment should also be considered before considering a compensation programme. Most of the implementing countries surveyed in this article have not assessed the positive impact of no-fault compensation programmes on their vaccination programmes. However, there is no published data that suggests a negative impact of vaccine injury compensation programmes on immunization programmes. Unlike previous publications that have placed emphasis on the protection of vaccine manufacturers from liability, sustaining vaccine supply and stabilizing vaccine prices as benefits of compensation programmes [1, 2, 6], our findings suggest that these programmes are increasingly being considered for fair compensation of injured party and maintain confidence in immunization programmes. This perception has the potential to encourage countries to implement compensation programmes

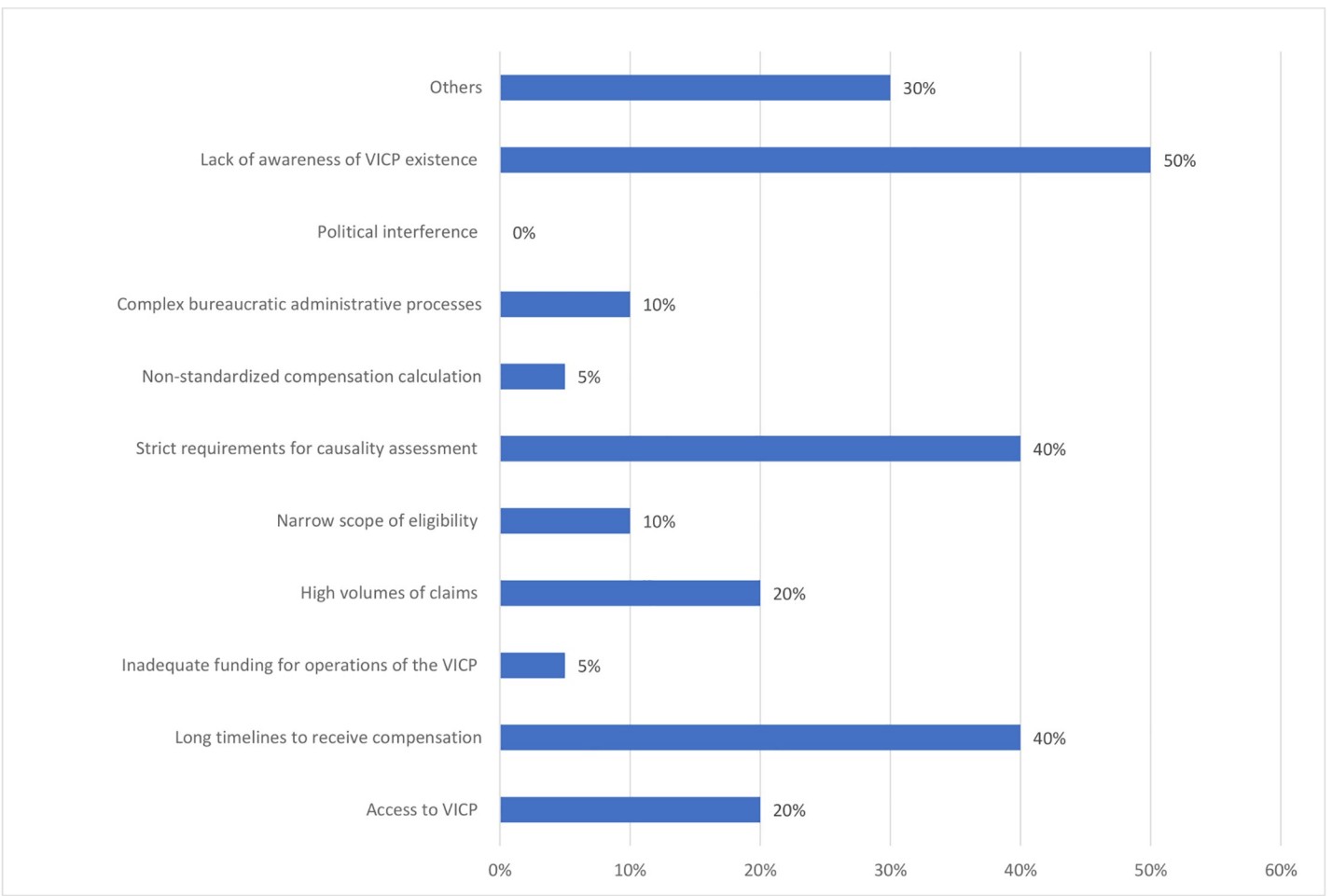

**Fig 4. Operational challenges of no-fault compensation programmes for vaccine injuries.**

for the benefit of the population and not perceived as merely protecting the interests of vaccine manufacturers.

From the identified operational challenges for compensation programmes, policy formulation should include clear and appropriate communication strategies to ensure public awareness of compensation procedures, while not raising undue concerns and confidence issues in vaccination. Efficient systems should focus on developing policies that would allow programmes to process claims within an acceptable turnaround time, reduce bureaucratic challenges, have standardized procedures to ensure equity and fairness, and have dedicated funding mechanism to ensure programme sustainability.

Despite not having data from all main survey respondents, our approach of triangulating information from multiple sources allowed us to have a general picture of 88% of the existing programmes. This study did ask national respondents to state how they implement their programme in a structured way. This enriches our findings as it provides first-hand information from reliable sources which may be missed by literature review and grey literature searches alone. Data on policies and practices of no-fault compensation programmes from Iceland, Russia, and Thailand was not available at the time of documenting the results. This resulted in some incompleteness of the current evaluation. However, as each of the programmes is implemented uniquely in the context of country economic capacity and legal systems, the available

information will be useful in guiding policy reviews and formulation for the next set of adopters. An important aspect to implementing VICP is understanding the main drivers for countries implementing compensation policies. Despite knowing motivations for implementing countries to adopt VICP policies, drivers for new implementers especially in less resourced settings remain undocumented. Our findings did not elaborate on this aspect and this remains an important area to explore as motivation for implementing VICP are likely to be different in varying socio-economic settings.

## Conclusion

As countries expand their use of vaccine and strengthen their vaccine safety surveillance and investigative capacity, occasional severe vaccine reactions are identified. Subsequently, the question of fair and equitable compensation of identified vaccine injuries is more frequently raised. Findings from this study demonstrate that interest in this issue is no longer limited to high-income countries. They also demonstrate the diversity of approaches that have been selected so far, thereby justifying the development of global guidance documents. The current absence of evidence related to the impact of such programmes on vaccine confidence and clarification on the purpose (justice, ethical requirement in case of mandatory vaccination, reduced litigations among others) will have to be clarified in elaborating such guidance.

## Disclaimer

The authors alone are responsible for the views expressed in this article and they do not necessarily represent the views, decisions or policies of the World Health Organization or of the other institutions with which they are affiliated.

## Supporting information

**S1 File. Search strategy_landscape-analysis.**
(DOCX)

**S2 File. Profile of survey respondents.**
(DOCX)

**S3 File. PRISMA-ScR checklist.**
(DOCX)

**S4 File. English questionnaire.**
(PDF)

**S5 File. French questionnaire.**
(PDF)

## Acknowledgments

We sincerely thank all the survey respondents of the study who took the time to provide us with useful information on how no-fault compensation programmes are implemented in their jurisdictions. Special thanks to the following colleagues who assisted in identifying experts to take part in our study; Oleg Benes, Dr Kari Johansen, Dr Eugene Lam, Dr Houda Langar, and Dr Shuyan Zuo. We sincerely acknowledge Professor Sue Ann Clemens, Dr Ralf Clemens, Dr Bernadette Hendrickx and the management team of the Master in Vaccinology and Pharmaceutical Clinical Development at the University of Siena for the support provided during this work.

## Author Contributions

**Conceptualization:** Randy G. Mungwira, Christine Guillard, Patrick L. F. Zuber.

**Data curation:** Randy G. Mungwira.

**Formal analysis:** Randy G. Mungwira, Patrick L. F. Zuber.

**Investigation:** Randy G. Mungwira, Patrick L. F. Zuber.

**Methodology:** Randy G. Mungwira, Christine Guillard, Adiela Saldaña, Nobuhiko Okabe, Helen Petousis-Harris, Edinam Agbenu, Lance Rodewald, Patrick L. F. Zuber.

**Project administration:** Randy G. Mungwira, Patrick L. F. Zuber.

**Supervision:** Christine Guillard, Adiela Saldaña, Nobuhiko Okabe, Helen Petousis-Harris, Edinam Agbenu, Lance Rodewald, Patrick L. F. Zuber.

**Validation:** Randy G. Mungwira, Christine Guillard, Patrick L. F. Zuber.

**Writing – original draft:** Randy G. Mungwira, Christine Guillard, Patrick L. F. Zuber.

**Writing – review & editing:** Randy G. Mungwira, Christine Guillard, Adiela Saldaña, Nobuhiko Okabe, Helen Petousis-Harris, Edinam Agbenu, Lance Rodewald, Patrick L. F. Zuber.

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
