## [Decision Letter · Decision Letter 0]

2 Dec 2019

PONE-D-19-20197

Manuscript Global landscape analysis of no-fault compensation programmes for vaccine injuries

PLOS ONE

Dear Dr Mungwira,

Thank you for submitting your manuscript to PLOS ONE. After careful consideration, we feel that it has merit but does not fully meet PLOS ONE’s publication criteria as it currently stands. Therefore, we invite you to submit a revised version of the manuscript that addresses the points raised during the review process.

We would appreciate receiving your revised manuscript by Dec 02 2019 11:59PM. To enhance the reproducibility of your results, we recommend that if applicable you deposit your laboratory protocols in protocols.io, where a protocol can be assigned its own identifier (DOI) such that it can be cited independently in the future. For instructions see: http://journals.plos.org/plosone/s/submission-guidelines#loc-laboratory-protocols

We look forward to receiving your revised manuscript.

Kind regards,

Holly Seale

Academic Editor

PLOS ONE

Journal Requirements:

2. Please include a separate caption for each figure in your manuscript.

3. We note that this manuscript reports a scoping review. While PLOS ONE does consider systematic and scoping reviews, in which authors address a clearly defined research question; conduct a systematic and comprehensive literature review; and use clearly reported, reproducible, and systematic methods to identify, select, and extract data from relevant research. We ask that when making revisions to your manuscript that you please include as Supporting Information, a PRISMA flow chart as your first figure and a PRISMA-ScR checklist (http://www.prisma-statement.org/Extensions/ScopingReviews). Please include all the information requested in this checklist in the methods section of your manuscript as well. If the Scoping Review has already been published, please provide a reference.

4.  We also ask that you include in your Methods section additional information about the professional participant recruitment method and the demographic details of your participants. Please ensure you have provided sufficient details to replicate the analyses such as:

a) the recruitment date range (month and year), 

b) a description of how participants were recruited, and

c) descriptions of where participants were recruited and where the research took place.

Reviewers' comments:

Reviewer's Responses to Questions

**Comments to the Author**

1. Is the manuscript technically sound, and do the data support the conclusions?

Reviewer #1: Yes

Reviewer #2: Yes

2. Has the statistical analysis been performed appropriately and rigorously? 

Reviewer #1: N/A

Reviewer #2: N/A

3. Have the authors made all data underlying the findings in their manuscript fully available?

Reviewer #1: Yes

Reviewer #2: Yes

4. Is the manuscript presented in an intelligible fashion and written in standard English?

Reviewer #1: Yes

Reviewer #2: Yes

5. Review Comments to the Author

Reviewer #1: This article reports the results of a multi-pronged research project to find out which WHO member states have NFC and how those schemes operate. The article is well-written, well-researched, makes a large contribution to knowledge and considers the key implications of NFC schemes. It should be published subject to the following issues being addressed.

There appear to be two broad ways that governments can set up NFC schemes – either having purpose built schemes, or incorporating vaccine injuries as part of other compensation schemes, as in New Zealand. Looker and Kelly, in their study, only appeared to have looked for specific standalone schemes. We did the same in our study (Attwell, K., Drislane, S., and Leask, J. 2019. Mandatory vaccination and no fault vaccine injury compensation schemes: An identification of country-level policies. Vaccine 37 (1):2483-2348. doi: https://doi.org/10.1016/j.vaccine.2019.03.065.) which sought to identify only whether countries with mandatory vaccination also had NFC. (And judging by your results, it looks like we missed Latvia’s NFC scheme, although it looks like a ‘covers NFC by other means’ policy.)

This current study contributes way beyond this by having asked countries what they do (rather than just searching in the public domain) and it is this that appears to have allowed researchers who lack language and local policy knowledge to actually find out how local policies operate that may be missed by database and grey lit searches. This differentiation, and hence contribution, should be emphasised in the intro and the discussion, and maybe even the abstract. If this is the first time such detailed and triangulated methods have been used to explore who has NFC, then that should be stated.

That said, the limitation noted at line 339 now raises some confusion. It seems like maybe you specifically looked for NFC, but if you found something else that did the same job, you also included it. It would be worth noting in more detail in your methods and results what happened in this regard. For example, did your ‘screening’ of countries ask them in a binary fashion, or allow them to tell you if they had something that did the job (eg new Zealand). Or did you find out about cases like NZ from other literature? I think it is important for readers to better understand your methodology, and hence the limtations, so you should elaborate these processes.

Line 97 ‘all WHO member states were screened’ – what does this mean? Are you introducing the steps you describe in the following sentences, or was this ‘screening’ something else? It sounds like an in introduction, so maybe make this clear eg. ‘were screened using several methods…’ It sounds later (results) like you actually contacted every member state as part of this screening. If this is true, then say so back here at line 97. If not, then the first part of the Results needs rewording.

Line 107 maybe insert ‘with a no-fault scheme identified through these machanisms…’ and then continue the sentence with what the survey was seeking to find out.

Line 121 onwards – was the 151 feedback simply countries telling you that they did or didn’t have NFC, or other things? Please specify this more clearly. Were surveys and other data sources used in conjunction in all of the 23 programs you evaluated, or just in some? What determined whether you did or didn’t use government docs?

Line 172 – since Japan doesn’t have mandatory vax, consider rewording the entire sentence or leave Japan out of it (and put in own category).

Line 298. This appears to be the first time you told us you asked the participants this question. It’s so good that you did. Please tell us earlier in the article how you did and why you did. Likewise for their assessments of challenges. To this point, I was expecting that you’d just followed Evans’ structure. Better and more expansive questions also enhances the contribution of this paper and should be emphasised elsewhere in the text.

Line 302-enhance legal basis for mandatory vax systems – presumably only if they had such systems in place.

Reviewer #2: 1. The methods section would benefit from further detail being added. At the moment it feels like a summary rather then a detailed (and replicable) outline of the processes used to collect and analyse the data.

2. The term 'cantonal' may not be understood by all readers

3. In the results section, it would be interesting to examine whether there are any common denominators amongst the policy components from countries that have introduced the vaccine compensation program in the last 10 years. Have they been influnced by the programs that have existed previously, by the published literature on issues arrising etc.

4. I am surprised that the challenges section is so succinct- I would have expected to see a richer outline here supported by the survey responses/quotes (if any open questions were used).

6. PLOS authors have the option to publish the peer review history of their article (what does this mean?). If published, this will include your full peer review and any attached files.

Reviewer #1: No

Reviewer #2: No

---

## [Author Response · Author response to Decision Letter 0]

23 Mar 2020

5. Review Comments to the Author

Reviewer #1: This article reports the results of a multi-pronged research project to find out which WHO member states have NFC and how those schemes operate. The article is well-written, well-researched, makes a large contribution to knowledge and considers the key implications of NFC schemes. It should be published subject to the following issues being addressed.

There appear to be two broad ways that governments can set up NFC schemes – either having purpose built schemes, or incorporating vaccine injuries as part of other compensation schemes, as in New Zealand. Looker and Kelly, in their study, only appeared to have looked for specific standalone schemes. We did the same in our study (Attwell, K., Drislane, S., and Leask, J. 2019. Mandatory vaccination and no fault vaccine injury compensation schemes: An identification of country-level policies. Vaccine 37 (1):2483-2348. doi: https://doi.org/10.1016/j.vaccine.2019.03.065.) which sought to identify only whether countries with mandatory vaccination also had NFC. (And judging by your results, it looks like we missed Latvia’s NFC scheme, although it looks like a ‘covers NFC by other means’ policy.)

This current study contributes way beyond this by having asked countries what they do (rather than just searching in the public domain) and it is this that appears to have allowed researchers who lack language and local policy knowledge to actually find out how local policies operate that may be missed by database and grey lit searches. This differentiation, and hence contribution, should be emphasised in the intro and the discussion, and maybe even the abstract. If this is the first time such detailed and triangulated methods have been used to explore who has NFC, then that should be stated.

Authors: Thank you for the detailed and thoughtful feedback on the manuscript. We agree with your observations on the importance and uniqueness of the multi-pronged approach to triangulate information which to our knowledge would be the first to be published. This has been clearly stated earlier in the introduction in the revised manuscript. We appreciate sharing your recent work on identifying country level policies with regards to mandatory vaccination and no-fault vaccine injury compensation, a great contribution to the pull of evidence to guide the next set of countries to adopt NFC. We accessed information from Latvia using a survey and information available from a Government website – the Treatment Risk Fund which covers injuries arising from vaccinations on a no-fault basis as other existing schemes. Once again thank you for the review and we have taken into consideration all your input in the revised manuscript. 

That said, the limitation noted at line 339 now raises some confusion. It seems like maybe you specifically looked for NFC, but if you found something else that did the same job, you also included it. It would be worth noting in more detail in your methods and results what happened in this regard. For example, did your ‘screening’ of countries ask them in a binary fashion, or allow them to tell you if they had something that did the job (e.g. New Zealand). Or did you find out about cases like NZ from other literature? I think it is important for readers to better understand your methodology, and hence the limitations, so you should elaborate these processes.

Authors: Thank you for this observation. We have clarified in the methods section how we approached countries to identify existing programmes. In summary, we approached each Member States EPI programme focal points as listed from WHO EPI team in HQ - Geneva, we also utilized experts in immunization programmes that we had come across in our professional engagements to identify suitable personnel who would respond (or find out in country from relevant authorities) to whether a country had an existing programme based on a preset definition of no-fault compensation programme. An initial email was sent out and if the respondent met the criteria set, a structured questionnaire collecting detailed information of the programme was shared with the relevant authorities within the compensation programme or the EPI programme.

Line 97 ‘all WHO member states were screened’ – what does this mean? Are you introducing the steps you describe in the following sentences, or was this ‘screening’ something else? It sounds like an in introduction, so maybe make this clear e.g. ‘were screened using several methods…’ It sounds later (results) like you contacted every member state as part of this screening. If this is true, then say so back here at line 97. If not, then the first part of the Results needs rewording.

Authors: Thank you for this observation. Indeed, we approached all Member States (MS). We complimented this effort with several other screening methods as described in the following sentences to identify existing n-fault compensation programmes particularly in instances where we did not receive any feedback. We have reworded as suggested for better clarity. 

Line 107 maybe insert ‘with a no-fault scheme identified through these mechanisms…’ and then continue the sentence with what the survey was seeking to find out.

Authors: This suggestion has been incorporated in the revised manuscript.

Line 121 onwards – was the 151 feedback simply countries telling you that they did or didn’t have NFC, or other things? Please specify this more clearly. Were surveys and other data sources used in conjunction in all of the 23 programs you evaluated, or just in some? What determined whether you did or didn’t use government docs?

Authors: Thank you for this observation. Please refer to figure 1 which provides a concise summary of how programme assessment and what sources were utilized. In summary, we approached each Member States EPI programme focal points as listed from WHO EPI team in HQ - Geneva, we also utilized experts in immunization programmes that we had come across in our professional engagements to identify suitable personnel who would respond (or find out in country from relevant authorities) to whether a country had an existing programme based on a preset definition of no-fault compensation programme. An initial email was sent out and if the respondent met the criteria set, a structured questionnaire collecting detailed information of the programme was shared with the relevant authorities within the compensation programme or the EPI programme. As this was a process of triangulating information, not all 23 programmes were evaluated based on survey and other data sources but rather a mixture of methods if more than one was available, including government documents if available on public domains or provided by the survey respondent. 

Line 172 – since Japan doesn’t have mandatory vax, consider rewording the entire sentence or leave Japan out of it (and put in own category).

Authors: Thank you for this observation. 

We understand that Japan has 2systems for immunization. !1. National Immunization Programme (NIP) and non-NIP vaccines. 

For NIP vaccines, these are listed in the law in two (2) categories: Type A vaccines which are proactively-recommended by Government (including most of childhood vaccines) meant to achieve heard immunity. This information and terminology of mandatory/pro-actively recommended by Govt. were confirmed with the Unit Chief Health Service Bureau Vaccination Team - Ministry of Health Labor and Welfare of Japan.

We have reworded the sentence for better clarity. 

Line 298. This appears to be the first time you told us you asked the participants this question. It’s so good that you did. Please tell us earlier in the article how you did and why you did. Likewise, for their assessments of challenges. To this point, I was expecting that you’d just followed Evans’ structure. Better and more expansive questions also enhances the contribution of this paper and should be emphasized elsewhere in the text.

Authors: Thank you for this observation. We have now stated earlier in the revised manuscript indicating we inquired of the benefits and challenges faced by existing no-fault compensation programmes for those who responded to the survey. 

Line 302-enhance legal basis for mandatory vax systems – presumably only if they had such systems in place.

Authors: This has been made clear in the revised manuscript.

Reviewer #2: 1. The methods section would benefit from further detail being added. At the moment, it feels like a summary rather than a detailed (and replicable) outline of the processes used to collect and analyze the data.

Authors: The methods section has been expanded to include further information as required by the PRISMA-ScR checklist. We have also clarified the outline of the process for the collection of data and its subsequent analysis in the revised manuscript.

2. The term 'cantonal' may not be understood by all readers

Authors: Thank you for this observation. We have clarified the term in the revised manuscript.

3. In the results section, it would be interesting to examine whether there are any common denominators amongst the policy components from countries that have introduced the vaccine compensation program in the last 10 years. Have they been influenced by the programs that have existed previously, by the published literature on issues arising etc.

Authors: We agree that this is an interesting area to examine if there are any similarities in recently introduced programmes. From our evaluation, we noted differences in how each programme is being implemented partially due to fundamental differences in each Member States laws guiding immunization and compensation. The authors efforts to achieve some way to compare existing programmes was by categorizing into six (6) elements as suggested by previous reviews whilst highlighting notable differences/uniqueness of programmes. 

4. I am surprised that the challenges section is so succinct- I would have expected to see a richer outline here supported by the survey responses/quotes (if any open questions were used).

Authors: Thank you for this observation. We had included open ended questions but did not get much information rather than what we thought would be challenges from the discussions we had experts in the field, review of observed challenges from previous publications. Also, important to note the challenge of getting information on this query from our targeted participants, mostly senior officials who may have a different perception of challenges with the programme compared to end-users.

---

## [Editor Report · Decision Letter 1]

5 May 2020

Global landscape analysis of no-fault compensation programmes for vaccine injuries: a scoping review and survey of implementing countries.

PONE-D-19-20197R1

Dear Dr. Mungwira,

We are pleased to inform you that your manuscript has been judged scientifically suitable for publication and will be formally accepted for publication once it complies with all outstanding technical requirements.

With kind regards,

Holly Seale

Academic Editor

PLOS ONE

Additional Editor Comments (optional):

The authors have updated the paper as per the reviewers suggestions
---

## [Editor Report · Acceptance letter]

8 May 2020

PONE-D-19-20197R1 

Global landscape analysis of no-fault compensation programmes for vaccine injuries: a scoping review and survey of implementing countries. 

Dear Dr. Mungwira:

I am pleased to inform you that your manuscript has been deemed suitable for publication in PLOS ONE. Congratulations! Your manuscript is now with our production department. 

With kind regards,

on behalf of

Dr. Holly Seale 

Academic Editor

PLOS ONE